# Fusion of GNSS Pseudoranges with UWB Ranges Based on Clustering and Weighted Least Squares

**DOI:** 10.3390/s23063303

**Published:** 2023-03-21

**Authors:** Günther Retscher, Daniel Kiss, Jelena Gabela

**Affiliations:** Department of Geodesy and Geoinformation, TU Wien—Vienna University of Technology, 1040 Vienna, Austria

**Keywords:** ultra-wideband (UWB), global navigation satellite system (GNSS), integration, sensor fusion, performance analysis

## Abstract

Global navigation satellite systems (GNSSs) and ultra-wideband (UWB) ranging are two central research topics in the field of positioning and navigation. In this study, a GNSS/UWB fusion method is investigated in GNSS-challenged environments or for the transition between outdoor and indoor environments. UWB augments the GNSS positioning solution in these environments. GNSS stop-and-go measurements were carried out simultaneously to UWB range observations within the network of grid points used for testing. The influence of UWB range measurements on the GNSS solution is examined with three weighted least squares (WLS) approaches. The first WLS variant relies solely on the UWB range measurements. The second approach includes a measurement model that utilizes GNSS only. The third model fuses both approaches into a single multi-sensor model. As part of the raw data evaluation, static GNSS observations processed with precise ephemerides were used to define the ground truth. In order to extract the grid test points from the collected raw data in the measured network, clustering methods were applied. A self-developed clustering approach extending density-based spatial clustering of applications with noise (DBSCAN) was employed for this purpose. The results of the GNSS/UWB fusion approach show an improvement in positioning performance compared to the UWB-only approach, in the range of a few centimeters to the decimeter level when grid points were placed within the area enclosed by the UWB anchor points. However, grid points outside this area indicated a decrease in accuracy in the range of about 90 cm. The precision generally remained within 5 cm for points located within the anchor points.

## 1. Introduction

Global navigation satellite systems (GNSSs) and ultra-wideband (UWB) are currently two central research topics when it comes to positioning in GNSS-denied/challenging environments. GNSS methods require a line-of-sight (LoS) between the satellite (transmitter) and GNSS-antenna (receiver) for the duration of the measurements. However, obstacles like tall buildings, trees and vehicles may disrupt the signal path and cause various undesirable effects. Therefore, UWB ranging is used in this study in order to mitigate the effects of such environments. In other words, the UWB measurements are used to augment GNSS positioning. UWBs have previously been tested for their performance in combination with GNSSs and have been proven useful in diverse cooperative positioning systems.

Initial research has already been performed by the authors of [1,2]. In these studies, the evaluation of raw data was performed with an extended Kalman filter (EKF). In comparison, this study focuses on a positioning solution of single points in relation to a set of anchor points in a GNSS-challenged environment. The coordinates of single points are evaluated with the statistical calculation method weighted least squares (WLS) using three different functional models (UWB-only, GNSS-only, GNSS/UWB-fusion). WLS is a calculation method that incorporates statistical and functional models in order to estimate the most statistically plausible values for the given models. The functional models are derived from the mathematical context of the point network built for the measurements. The network of points was built consisting of a set of four anchor points with known coordinates and seven so-called grid test points (i.e., single points) that were chosen to be in a GNSS-friendly and GNSS-challenged environment. Another criterion for the selection of grid points was their location inside and outside the area enclosed by the anchor points. For the evaluation of the collected GNSS raw data, the open-source software package RTKLib was used. The following steps of the evaluation, such as data preparation (filtering and assignment), implementation of the WLS algorithm, and calculating statistical values for the final evaluation were executed in MATLAB.

The major aim of this study was to investigate and compare the positioning solutions of the three implemented WLS approaches. Furthermore, we tested whether the measurement and evaluation methods used were practicable and whether they should be used in further research.

The paper is organized as follows: Section 2 presents a general overview of the two positioning methods, GNSS and UWB, including how they operate and their possible combinations and comparable measurement methods. In Section 3 the methodology of data processing is thoroughly described, while Section 4 details how the field test was carried out and how the raw data were collected. Section 5 deals with the visualization and interpretation of the major results as well as what conclusions can be drawn from them. Finally, Section 6 concludes the paper and provides an outlook on future developments.

## 2. Employed Positioning Technologies

In this study, the usage of GNSS and UWB is investigated either in standalone mode or integrated to an ubiquitous positioning solution. Measurements were taken in a test field to verify their performance. In the following, the two positioning methods are briefly reviewed.

### 2.1. Global Navigation Satellite Systems (GNSSs)

GNSS is short for global navigation satellite system, which includes the US Navstar GPS (Global Positioning Service), the Russian GLONASS (Global’naya Navigatsionnaya Sputnikkovaya Sistema), the European Galileo, and the Chinese BeiDou Navigation Satellite System (BDS) constellations [3]. Each of these systems consists of multiple satellites in different satellite orbits. In order to determine precise and accurate coordinates on Earth, at least four satellites are needed. Satellite orbits or ephemerides are known, and each satellite uses high-precision atomic clocks to determine time.

The basic principle of GNSS positioning works as follows [4]: each of the satellites broadcasts code and carrier frequency signals to the receiver on Earth and the travel time of the signal is measured. With the measured travel time from the code signals, so-called `pseudoranges’ can be determined. Due to the asynchrony of the satellite and receiver clocks’ travel time, measurements are faulty and must be corrected. The receiver clock error can be estimated when at least four satellites are available and tracked. The clock error is then included in the calculation of the distances between the satellite and the receiver, to determine the pseudoranges. The position of the receiver on Earth is then estimated using a 3D intersection based on the multi-lateration concept.

In order to estimate the position of the receiver on Earth with high precision, so-called `carrier phases’ (CPs) are used. In combination with the earlier described code pseudoranges, it is possible to achieve low measurement noise and with that, a precision in the range of a few millimeters or centimeters with geodetic GNSS equipment. To achieve this result, it is important to include various effects in the observation equation. Such effects include clock errors, propagation effects of the atmosphere (troposphere and ionosphere), multi-path effects, satellite orbit errors, relativistic effects and most importantly carrier phase ambiguities. Most of these effects can be reduced or even eliminated using single, double or triple differences (SDs, DDs, TDs). Single differences can be calculated when there are two simultaneous measurements of two GNSS receivers from one satellite. By using this method, satellite clock errors, orbit errors, ionospheric effects and tropospheric travel time delays can be reduced. By using double differences that are available when two receivers simultaneously measure to two satellites, receiver clock error and phase offsets of the receiver can be eliminated. Triple differences can be calculated using simultaneous measurements of two receivers from two satellites at different time epochs. When processing DDs, the number of whole wavelengths between receiver and satellite must be estimated because they are essential in precise positioning. The number of whole wavelengths is called the ambiguity. These ambiguities can be estimated by implementing special methods using the least-squares method, e.g., the least-squares ambiguity decorrelation adjustment (LAMBDA) method [5]. Since signals are emitted as electromagnetic waves by satellites, they travel through different layers of the atmosphere, which causes the signal path to be redirected. In order to address these atmospheric effects, the atmosphere is separated into a tropospheric and an ionospheric portion [4]. By modeling both atmospheric sections, the signal path can be corrected. Additionally, there are a few types of interference that affect these signals on Earth and weaken the signal. The amount by which the signal is weakened depends on the properties of the materials and the geometry (e.g., refraction). One major interference is the multi-path effect. Signals might face obstacles in their path and are often obstructed and reflected by buildings, trees, or other large objects [6]. Because of reflection and diffraction, the signals can find their way to the receiver in multiple ways. The combination of direct and indirect signals causes interference in the raw data. In order to reduce these effects, choke-ring antennas or multi-path reducing tracking loops in the receiver can be used. Longer observation times and time-stacking methods help to reduce multi-path effects as well [7]. The mentioned obstacles can also influence the satellite visibility due to the signal’s inability to pass through obstacles. Due to losses of satellite availability, positioning solutions get worse the smaller the number of satellites is. Satellites with low elevation are more likely to be affected by the obstructions.

As a GNSS does not work in all environments, alternatives are needed to provide ubiquitous positioning solutions. UWB can be such an alternative and has been chosen in this work as an alternative method either in standalone mode or fused with a GNSS. The fusion of technologies is promising and can be the way to achieve ubiquitous positioning. The basics of UWB technology are discussed in the next section.

### 2.2. Ultra-Wide Band (UWB)

UWB ranging is a ranging method used in this work to support GNSS measurements in challenging environments, such as in GNSS signal-obstructed environments as well as in transitioning indoors/outdoors environments. UWB works with a bandwidth of radio frequencies greater than 500 MHz and is usually operated in a range of 3.1 to 10.6 GHz [8,9]. Due to its high bandwidth, UWB is robust to interference caused by multiple paths [10]. The lower frequencies support the penetration through obstacles, which enables the multi-path resolution capabilities in environments that are disadvantageous for GNSS measurements. Another advantage of such high bandwidth is a very fine time-resolution, which is about hundreds of picoseconds. Converted to spatial resolution, these hundreds of picoseconds result in centimeter-level accuracies. With that fine resolution time, range observations can be easily separated from measurements that suffer from multi-path effects. UWB devices are able to send out weak signals that support the feature of not having an impact on other systems that have the same scale in bandwidth, such as Wi-Fi (Wireless-Fidelity) [11,12]. The low-cost method of construction of UWB equipment and its functionality as transceivers with long battery run-times are also factors in favor of its usage [1].

To be able to achieve the scalability properties, synchronization and power control, certain measures need to be implemented. The main problems to be faced are that there are strict requirements on how antennas should be built regarding their size and shape. Challenging conditions in the surroundings of the UWB device must be addressed properly, since the transceiver must be able to work in a wide frequency band in varying environmental conditions. Generally, UWB ranges can be measured up to about 200 m in line-of-sight (LoS) environments but the performance for ranging is best when range observations are performed under 50 m [13].

UWB measurements can be grouped into two categories, fingerprinting-based and geometric methods. As part of this work, ranging based on the latter method was used. Geometric methods include range and angle information gathering derived from received signal strength indicator (RSSI), time-of-arrival (ToA), time-difference-of-arrival (TDoA) or angle-of-arrival (AoA) measurements observed through a UWB system (see, e.g., [1,14,15]).

In the context of this study, transmitters and receivers based on systems that utilize measurements of ’two-way time-of-flight’ (TW-ToF) were used [11]. As this method implies, the time of the signal between transmitter and receiver $t_{1}$, as well as the time on the way back from the receiver to the transmitter $t_{2}$, including the time it takes the receiver to send out the signal again, are observed. Hence, TW-ToF measures the double ranges $r_{obs}$ from the transmitter to the receiver $t_{1}$ and back to the transmitter $t_{2}$. It also measures the time taken by the receiver to respond to the transmitter UWB $t_{d}$ to derive the range observations between a transmitter and receiver. The time $t_{d}$ is generally constant and can be estimated using calibration. Then, the range observation $r_{obs}$ is obtained as given in Equation (Equation 1) [11]:(1)$$r_{obs}=\frac{1}{2}*\left(\left(t_{1}+t_{2}+t_{d}\right)-t_{d,calib}\right)*c$$
where *c* is the propagation speed of the radio frequency (RF) signal and $t_{d,calib}$ is the estimated value of $t_{d}$ obtained after calibration [1].

UWB units utilize coherent (i.e., phase-stable continuous oscillation) transmission of very-short-duration RF wave-forms (referred to as pulses). Packets of several thousands of these short pulses are then transmitted for estimating the required travel time for the RF signal between UWB nodes. Accurate detection of the first pulse (first break) or leading edge (LE) is then utilized for the range measurement of the direct signal. At the same time, multipath and NLoS effects are filtered out [16]. Time synchronization of the transmitting and receiving devices is a substantial requirement that is usually achieved through the UWB hardware. A master UWB unit is defined for that purpose. Due to the coherent transmission capabilities, and through implementing the TW-ToF technique, synchronization issues are resolved to a great extent [11]. This enables range measurements at centimeter-level accuracy.

Using this method, positions can be calculated using multilateration. Since the transmitter and receiver times are not synchronized, it is mandatory that when the signal is sent by the transmitter to the receiver, it is sent back as well. As aforementioned, UWB sensors transmit short-duration RF wave-forms. One transmission is made from several thousands of pulses. These pulses enable high-accuracy range measurements, which have high practicability in multipath environments with either line-of-sight (LoS) or non-line-of-sight (NLoS) conditions.

The nominal accuracy was validated in [17] for the order of around 3 cm for calibrated UWB pairs. Two different off-the-shelf UWB systems (the P410 and P440 TimeDomain [18] and the Poxyz [19] UWB modules) were tested in an indoor setting along a hallway in [20]. For these tests, 14 TimeDomain and 14 Pozyx UWB anchors (also referred to as static nodes) were mounted on the walls along the building corridor. Four mobile rover units were used by pedestrian users walking at typical speed or in stop-and-go mode along the hallway with 35 checkpoints. The experiments indicated that calibration of the UWB units and derived ranges is essential to achieve a high level of performance and positioning accuracies at the cm-level [11]. UWB for the navigation of visually impaired people was employed in [21]. They found that UWB technology is very useful for such applications navigating in 2D in large buildings. The use of UWB for navigation of robots is discussed in [22]. To perform this task, a distributed SLAM (simultaneous localization and mapping) solution was employed to estimate the trajectory of a group of robots using UWB ranging and odometry measurements. The approach determines the relative pose (also known as loop closure) between two robots by minimizing the UWB ranging measurements taken at different positions when the robots are in close proximity. The authors of that study claimed that UWB provides a good distance measure in LoS conditions, but retrieving a precise pose estimation remains a challenge, due to ranging noise and the unpredictable path traveled by the robot.

Due to the development of low-cost UWB chip sets at small sizes, they have found their way into smartphones, leading to more widespread applications of UWB technology. Further discussion about UWB-enabled mobile devices will follow in the outlook in Section 6.

## 3. Multi-Sensor Positioning

This section describes the methodology used to process the GNSS and UWB data and to determine the user’s position. In order to extract the grid test points from the collected raw data, a clustering method was used (Section 3.1). The user’s static position was then determined using the weighted least squares (WLS) method. Section 3.2 defines WLS and provides functional models for a positioning solution based on UWB measurements only, GNSS measurements only and combined UWB and GNSS measurements (multi-sensor solution).

### 3.1. Clustering Methods

Generally, clustering is used in large spatial databases to determine areas where points occur in a concentrated form and are separated from each other by empty spaces. Clustering algorithms are mainly used for machine learning and to detect patterns in data. This section will introduce two clustering methods, an already existing method, density-based spatial clustering of applications with noise (DBSCAN) [23,24,25], and a self-written clustering method that is a derivation of DBSCAN. The self-written clustering method was used for raw data processing. Normally, it is necessary to note down the start and end times of the measurements of each point during field work. The clustering method is a substitution for this process that should reduce the number of tasks in the field.

#### 3.1.1. Density-Based Spatial Clustering of Applications with Noise (DBSCAN)

Density-based clustering is a method that sorts a set of data points by their spatial connectivity as well as their spatial density. With this method, it is possible to create three-dimensional connected objects in any form possible. An example of this situation could be districts that should be mapped according to their togetherness. One such method is DBSCAN [26]. This method identifies clusters in large spatial datasets by looking at the local density of the data points. There are two input parameters. The first parameter is epsilon (i.e., ϵ), which defines the radius of the circle created around each data point to check the density. The second parameter is MinPts. This is a parameter that denotes the minimum number of data points that are needed inside the circle for that data point to be assigned to be a so-called “Core-Point”. First, DBSCAN creates a circle around each data point with the radius ϵ and classifies them as either of three types of points (shown in Figure 1).

A data point is a Core Point if a circle around it includes at least the number of MinPts. A Border Point is defined as a directly reachable point from a Core Point within the radius ϵ. If a data point cannot be reached from any other point within the dataset, this point is called a Noise Point. Unlike other algorithms, DBSCAN only needs to run through the whole dataset once to complete the clustering.

There are two concepts that are the basis for such density-based clustering methods (shown in Figure 2). The first refers to situations where a data point is “density-reachable” when there is a set of points so that each point $P_{i+1}$ is directly reachable from $P_{i}$. The second concept addresses situations where two data points are reachable from a specific point regarding radius ϵ and minPts. These two data points are called “density-connected”. A density-connected cluster is built when there are points in the dataset that are density-reachable from any point in the dataset and when all points in the cluster are density-connected [23,24,25].

Core Points;Border Points; andNoise.

The process of clustering points in a large dataset is summarized in the flowchart in Figure 3.

DBSCAN is, as mentioned, a density-based clustering method. It is often used in situations where close togetherness is interesting. Prime examples of such situations are searching processes for areas where, for example, many people like to choose the location for a new restaurant or for areas where many accidents happen, to determine an appropriate location for a new hospital. It is also an extremely time-efficient method, which was discussed in Ester et al. [27].

#### 3.1.2. DBSCAN-Derived Clustering Method

The implemented clustering method used to filter the observed data is a derivative of DBSCAN. There are two main differences between those two implementations. The first is the number of parameters that are needed as input, which is three compared to the two of DBSCAN. The first two parameters define the maximum difference between two data points. The first parameter is used for broadly filtering the data and the second one takes these broadly filtered data points and calculates a fine-filtered solution. The third parameter has the same functionality as the minPts parameter of DBSCAN. The summary of the workflow of the DBSCAN-derived clustering method is shown in Figure 4. The second major difference lies in how the calculation of the density between points is implemented. As mentioned above, DBSCAN defines a circle and connects the points inside this circle to a cluster. The DBSCAN-derived clustering method calculates the differences between two successive three-dimensional data points and checks whether their differences are larger or smaller than the first input parameter for broad filtering. If the difference is smaller than the parameter, the data point is then added to a temporary dataset. If the difference is larger than the parameter for broad filtering and if the size of the temporary dataset is larger than zero, it will be checked if the temporary dataset contains more than the minimum number of points. If this condition applies, the mean value of these data points inside the temporary dataset will be estimated. This is comparable to the core points of DBSCAN. Additionally, the indices of the input dataset are saved where the clusters were found. If this condition does not apply, the temporary dataset will be cleared of all values. The second part of this function is then analog to the first part; the only differences are that the broad-filtered clusters are used as the input dataset and the second input variable is used to check the conditions of the data point differences. It is important to note that it is not necessary to compare the condition of the clusters with the minPts variable for the fine-clustering part because the broad-filtered clusters that are used as input already fulfill this condition. The final output values of this function are the first and last indices of the finalized clusters of the original dataset.

With the DBSCAN-derived method, it is possible to differentiate the coordinates of points that were collected as raw data. Figure 5 and Figure 6 show the observed raw data that were clustered using the DBSCAN and DBSCAN-derived method, respectively. The clusters are marked in different colors and each cluster has its own number, which is shown in the legend on the right side. Both methods were applied to the same dataset. The radius parameter ϵ for DBSCAN and the diff1 parameter were set to be the same value (ϵ=diff1=0.03). The minPts parameters for both methods were also chosen to be the same (minPts = 3). The results of DBSCAN clustering show 36 cluster classes in total and one class, indicated with −1 in the legend, that refers to noise points that could not be assigned to a cluster. The DBSCAN-derived method returned only 13 cluster classes and one that is marked with −1 for noise points. The results in Figure 5 show similar results compared to Figure 6. The number of DBSCAN clusters is higher compared to the DBSCAN-derived clustering method. The DBSCAN-derived method delivers a more useful result regarding the measurement campaign conducted as part of this work because it returns clusters as they were observed with the passage of time. That means that data points with similar coordinates might not belong to the same cluster due to different measurement times. Another reason in favor of the DBSCAN-derived method is that the number of clusters in the output is closer to the actual number of observed points in the field than with DBSCAN. A drawback of this function, and for density-based clustering methods in general, is that all clusters show some errors that refer to edge cases that could possibly belong to multiple clusters. After applying the method to the dataset, manual corrections must be made since some data points are always incorrectly assigned to clusters. The reason for this lies mostly in the choice of appropriate input parameters.

### 3.2. Weighted Least Squares (WLS)

After the data filtering and clustering are finished, the next step is the estimation of the ranges and coordinates. For this, the WLS method was chosen. The WLS approach is an estimation method that incorporates statistical models in order to estimate the most statistically plausible values. The first input parameter is the random vector of the observations *L* under the assumption of constant error variance. Additionally, the standard deviations $\sigma_{i}$ associated with the observations vector *L* are given.

#### 3.2.1. Functional Model

For the estimation, a functional model that incorporates the observations is needed. This model is realized through equations $\varphi_{i}$ that combine the observations $L_{i}$ with i=1,2,⋯,n and the unknown values $X_{j}$ for j=1,2,⋯,m as shown in:(2)L=φ(X)

To solve the equations, they must be linearized. The linearization is performed via differentiation of the equation after the unknown values $X_{j}$. Then, the functional model is derived with the model matrix *A*:(3)$$A=\left[\begin{array}{cccc} \frac{\partial \varphi_{1}}{\partial X_{1}} & \frac{\partial \varphi_{1}}{\partial X_{2}} & \cdots & \frac{\partial \varphi_{1}}{\partial X_{m}}\\ \frac{\partial \varphi_{2}}{\partial X_{1}} & \frac{\partial \varphi_{2}}{\partial X_{2}} & \cdots & \frac{\partial \varphi_{2}}{\partial X_{m}}\\ \vdots & \vdots & \ddots & \vdots \\ \frac{\partial \varphi_{n}}{\partial X_{1}} & \frac{\partial \varphi_{n}}{\partial X_{2}} & \cdots & \frac{\partial \varphi_{n}}{\partial X_{m}} \end{array}\right]$$

Section 3.3 details three ways the functional model and model matrix *A* will be set up in this paper.

The next step is to calculate the approximated observation vector $L_{0}$. This is executed by applying Equation (Equation 2). After that it is possible to calculate the shortened observation vector *l* with:(4)$$l=L-L_{0}$$

Furthermore, an approximated unknown vector $x_{0}$ is needed. The values in this vector can be approximated using various kinds of methods such as basic geodetic triangulation (e.g., arch section, forward resection, backward resection); GNSS measurements could also be used.

#### 3.2.2. Stochastic Model: A Priori

After the functional model is defined, a stochastic model can be set up. This depends on whether all the observation’s standard deviations are the same or whether they are all different. If all standard deviations are the same, it can be omitted. The stochastic model mainly consists of three major matrices:the observation’s variance-covariance matrix $\Sigma_{LL}$;the observation’s cofactor matrix $Q_{LL}$; andthe weights matrix *P*.

The first matrix to be built is the variance-covariance matrix $\Sigma_{LL}$. It consists of the observation variances on the main diagonal. Covariances are only needed when there is a correlation between the observations. If there is no correlation or it is assumed that there is no correlation, the matrix is diagonal (i.e., correlation elements are zero). In Equation (Equation 5), the matrix includes the calculations of covariances for completeness. It is important to note that the matrix is a square matrix with dimensions of n×n corresponding to the number of observations *n*.
(5)$$\Sigma {}_{LL}=\left[\begin{array}{cccc} \sigma {}_{11}^{2} & \sigma {}_{12} & \cdots & \sigma {}_{1n}\\ \sigma {}_{21} & \sigma {}_{22}^{2} & \cdots & \sigma {}_{2n}\\ \vdots & \vdots & \ddots & \vdots \\ \sigma {}_{n1} & \sigma {}_{n2} & \cdots & \sigma {}_{nn}^{2} \end{array}\right],$$

The cross-covariances are calculated from the product of the standard deviations $\sigma_{ii}$ with i=1,2,⋯,n of the two correlating observations and the correlation factor ρ. An example of this calculation is shown in Equation (Equation 6). It is important to note that covariances with the same but mirrored indices have the same value.
(6)$$\sigma {}_{12}=\sigma {}_{11}*\sigma {}_{22}*\rho$$

The second matrix (i.e., $Q_{LL}$) is the cofactor matrix that is calculated by the multiplication of a constant value with $\Sigma_{LL}$. This constant value is the variance of unitary weight $\sigma_{0}^{2}$ that can be freely chosen. This leads to:(7)$$Q_{LL}=\frac{1}{\sigma_{0}^{2}}\Sigma {}_{LL}$$

The last matrix is the weights matrix *P* and it is the inverse of the cofactor matrix $Q_{LL}$.
(8)$$P=Q_{LL}^{-1}$$

#### 3.2.3. Adjusted Observations and Unknown Values

With all mentioned vectors and matrices, the adjusted observations and unknown values can be estimated. A support matrix is needed: the matrix of normal equations *N* that is calculated as shown in Equation (Equation 9). The matrix $A^{T}$ denotes the transposed model matrix *A*:(9)$$N=A^{T}PA$$

The following equations lead to the results. With Equation (Equation 10) the correction vector *x* of the unknown values is estimated.
(10)$$x=N^{-1}A^{T}Pl$$

Additionally, Equation (Equation 11) estimates the corrections for the observations:(11)v=Ax−l

With these equations, all the required corrections can be calculated. The estimations of the adjusted observations $\hat{L}$ and the adjusted unknown values $\hat{X}$ are shown in Equations (Equation 12) and (Equation 13), respectively.
(12)$$\hat{L}=L+v$$
(13)$$\hat{X}=X_{0}+x$$

It is mandatory to execute a test if the estimated values have the required precision for the task. The estimated observations $\hat{L}$ and the estimated unknown values $\hat{X}$ are put into the functionality model. The result of this test should be close to zero. If the result of the test is not satisfactory, an iterative evaluation of the adjusted observations and unknown values has to be performed. The adjusted unknown values are set to be the approximated unknowns. The observation vector *L* and the stochastic model will stay the same. The process of the WLS approach is summarized in the flow chart in Figure 7.

### 3.3. Functional Model Definition

In this study, the coordinates of grid test points located within a set of anchor points are estimated. In this section, an example is given for one grid point *N*. In total there are four anchor points ($P_{1}$, $P_{2}$, $P_{3}$ and $P_{4}$) and one grid point (*N*) in between these anchor points (see Figure 8). Range measurements between each anchor point and the grid point *N* are $d_{1N}$, $d_{2N}$, $d_{3N}$ and $d_{4N}$.

To be able to compare the GNSS and UWB performance, the grid point coordinates are estimated for UWB only, GNSS only and a combination of the UWB and GNSS measurements. All coordinates are expressed in a metric Cartesian coordinate system.

#### 3.3.1. UWB-Only Model

It is assumed that the coordinates of the anchor points are known, the measured grid point *N* is only an approximation, and the UWB ranges are the observations comprising the observation vector *L*.

The first step is to set up the functional model. For this step, an equation that combines all values of the measurements is necessary. The equation in this situation is the distance formula as given in:(14)$$f:d_{i}=\sqrt{{\left(x_{N}-x_{i}\right)}^{2}+{\left(y_{N}-y_{i}\right)}^{2}+{\left(z_{N}-z_{i}\right)}^{2}}$$

Since there are four range measurements from four anchor points, there are also four equations. The non-linear functional model equations need to be linearized. The linearization is performed by finding differential derivatives of Equation (Equation 14) after the 3D coordinates of the grid point *N* as shown in Equation (Equation 15) for the *x*-coordinate only. The equations for the *y* and *z* coordinates are similar and only the index for the respective coordinates has to be used.
(15)$$\frac{\partial f_{i}}{\partial X_{n}}=\frac{x_{N}-x_{i}}{\sqrt{{\left(x_{N}-x_{i}\right)}^{2}+{\left(y_{N}-y_{i}\right)}^{2}+{\left(z_{N}-z_{i}\right)}^{2}}}=\frac{x_{N}-x_{i}}{d_{i}}$$

The values for $d_{i}$ in this case are the UWB range measurements. With these, the model matrix *A* has four rows since there are four equations, and three columns since there are three unknown values (i.e., the Cartesian coordinates (*x*, *y*, *z*) of the grid point). The model matrix *A* can then be written as in Equation (Equation 16).
(16)$$A_{\left(4\times 3\right)}=\left[\begin{array}{ccc} \frac{x_{N}-x_{1}}{d_{1}} & \frac{y_{N}-y_{1}}{d_{1}} & \frac{z_{N}-z_{1}}{d_{1}}\\ \frac{x_{N}-x_{2}}{d_{2}} & \frac{y_{N}-y_{2}}{d_{2}} & \frac{z_{N}-z_{2}}{d_{2}}\\ \frac{x_{N}-x_{3}}{d_{3}} & \frac{y_{N}-y_{3}}{d_{3}} & \frac{z_{N}-z_{3}}{d_{3}}\\ \frac{x_{N}-x_{4}}{d_{4}} & \frac{y_{N}-y_{4}}{d_{4}} & \frac{z_{N}-z_{4}}{d_{4}} \end{array}\right]$$

As explained in Section 3.2.1, the vector for the estimated observations $L_{0}$ can now be calculated. For this step, the anchor point coordinates and the grid point coordinates are the input parameters for the functional model’s equation (Equation (Equation 14)). Then, the shortened observations vector *l* can be calculated (Equation (Equation 4)). The last step of the derivation of the functional model is an approximation of the true values of the grid point coordinates $x_{0}$, which are determined as the mean values of all grid point observations. These steps are also shown in Figure 7 and explained in the previous sections.

#### 3.3.2. GNSS-Only Model

As before, in this section, it is assumed that the coordinates of the anchor points are known and the measured grid points are only an approximation. However, the observations for the observations vector *L* are calculated from baselines $s_{i}$ between the approximate grid point coordinates given by the GNSS receiver and the known anchor point coordinates.

The first step is to calculate the GNSS baselines $s_{i}$ between UWB anchor and grid points. As the anchor points are surveyed with static GNSS observations and therefore precisely known, they can be used for the baseline estimation. For the grid point *N*, the measurements during the test for each epoch are used. The equation used for the functional model is similar to Equation (Equation 14). The only difference is that the UWB range measurements $d_{i}$ are replaced by the calculated baselines $s_{i}$.
(17)$$\begin{array}{c} g:s_{i}=\sqrt{{\left(x_{N}-x_{i}\right)}^{2}+{\left(y_{N}-y_{i}\right)}^{2}+{\left(z_{N}-z_{i}\right)}^{2}} \end{array}$$

Their linerarization for the *x* coordinate results again in:(18)$$\begin{array}{c} \frac{\partial g_{i}}{\partial X_{n}}=\frac{x_{N}-x_{i}}{\sqrt{{\left(x_{N}-x_{i}\right)}^{2}+{\left(y_{N}-y_{i}\right)}^{2}+{\left(z_{N}-z_{i}\right)}^{2}}}=\frac{x_{N}-x_{i}}{s_{i}} \end{array}$$

The model matrix *A* can now be written as:(19)$$A_{\left(4\times 3\right)}=\left[\begin{array}{ccc} \frac{x_{N}-x_{1}}{s_{1}} & \frac{y_{N}-y_{1}}{s_{1}} & \frac{z_{N}-z_{1}}{s_{1}}\\ \frac{x_{N}-x_{2}}{s_{2}} & \frac{y_{N}-y_{2}}{s_{2}} & \frac{z_{N}-z_{2}}{s_{2}}\\ \frac{x_{N}-x_{3}}{s_{3}} & \frac{y_{N}-y_{3}}{s_{3}} & \frac{z_{N}-z_{3}}{s_{3}}\\ \frac{x_{N}-x_{4}}{s_{4}} & \frac{y_{N}-y_{4}}{s_{4}} & \frac{z_{N}-z_{4}}{s_{4}} \end{array}\right]$$

The following processing steps are then again as in Figure 7.

#### 3.3.3. GNSS/UWB Fusion Model

This section defines GNSS/UWB fusion. Sensor fusion of GNSS and UWB measurements is ensured by combining the two previous models and vectors such as observation vector *L* and model matrix *A*. Both Equations (Equation 14) and (Equation 17) are used as the functional model that needs to be linearized and for the derivation of the model matrix *A*. The linearization is performed as in Equations (Equation 15) and (Equation 18).

In order to set up the combination of GNSS and UWB, both the linearization of the UWB solution and the GNSS solution must be included in the model matrix *A* (see Equation (Equation 20)). Since there are now eight equations in total, the model matrix consists of eight rows, and the number of columns remains the same since the unknown variables remain unchanged. The upper four rows of Equation (Equation 20) contain the linearization of the equations for the UWB part of the solution, and the lower four rows are representative of the GNSS part of the model.
(20)$$A_{\left(8\times 3\right)}=\left[\begin{array}{ccc} \frac{x_{N}-x_{1}}{d_{1}} & \frac{y_{N}-y_{1}}{d_{1}} & \frac{z_{N}-z_{1}}{d_{1}}\\ \frac{x_{N}-x_{2}}{d_{2}} & \frac{y_{N}-y_{2}}{d_{2}} & \frac{z_{N}-z_{2}}{d_{2}}\\ \frac{x_{N}-x_{3}}{d_{3}} & \frac{y_{N}-y_{3}}{d_{3}} & \frac{z_{N}-z_{3}}{d_{3}}\\ \frac{x_{N}-x_{4}}{d_{4}} & \frac{y_{N}-y_{4}}{d_{4}} & \frac{z_{N}-z_{4}}{d_{4}}\\ \frac{x_{N}-x_{1}}{s_{1}} & \frac{y_{N}-y_{1}}{s_{1}} & \frac{z_{N}-z_{1}}{s_{1}}\\ \frac{x_{N}-x_{2}}{s_{2}} & \frac{y_{N}-y_{2}}{s_{2}} & \frac{z_{N}-z_{2}}{s_{2}}\\ \frac{x_{N}-x_{3}}{s_{3}} & \frac{y_{N}-y_{3}}{s_{3}} & \frac{z_{N}-z_{3}}{s_{3}}\\ \frac{x_{N}-x_{4}}{s_{4}} & \frac{y_{N}-y_{4}}{s_{4}} & \frac{z_{N}-z_{4}}{s_{4}} \end{array}\right]$$

## 4. Data Collection Campaign

In order to assess the positioning performance of GNSS in challenging environments when UWB range observations are included, real-world data were collected. This section details the survey area where open-sky and GNSS-challenged environments are available and where pedestrians are walking around and potentially affect the UWB measurement quality. The following parts of this section also detail the equipment and sensors used, the post-processing software used to evaluate the observed raw data and how the ground truth was determined.

### 4.1. Survey Area and Scenario

The chosen survey area is a small section of the Resselpark at Karlsplatz square in the fourth district of Vienna. The survey area contains open-sky areas, full-grown trees and a pond with a statue. Four anchor points (P19, P20, P07’ and P18’) were set up. Two anchor points are from the existing control point network (P19 and P20) and the other two were newly set up. A set of seven so-called grid points (G1–G7) were also set up. Most of these points are located within the area that the anchor points enclose. Figure 9 illustrates the distribution of anchor and grid points. The full extent of the survey area is about 630 m^2^ and with a maximum distance between two anchor points of about 54 m.

The measurements were carried out on a windy day at the beginning of April when a moderate number of pedestrians were present. GNSS data and UWB data were collected simultaneously. Each anchor point was equipped with one UWB TimeDomain unit fixed on a tripod.

The GNSS/UWB grid point measurements were taken in stop-and-go mode with a stop at each point for 2 to 3 min. The GNSS data collection was carried out with a sampling rate of one measurement per second, while the sampling rate for the UWB ranging was about 0.1 s. A dual-frequency surveying-grade GNSS receiver was fixed on a pole that also carried the UWB unit. A user then walked to each grid point and stopped (i.e., stop-and-go mode). Figure 10 shows the measurement network with the blue dots representing the grid points and the black triangles referring to the anchor points. The different colors of the ranges describe the ranges to a specific grid point (e.g., red for grid point G1).

The anchor point coordinates were determined by post-processing the static GNSS measurements, collected over a duration of about 15 min, with the precise ephemeris data (final orbits) from the International GNSS Service (IGS). Section 4.4 provides more information on this.

### 4.2. Sensors

Two sensor types were used in this data collection: UWB units and GNSS receivers. The GNSS observations were performed with two Spectra SP80 receivers capable of measuring at two carrier frequencies (L1 and L2 in the case of GPS). Five UWB units were needed to make all UWB measurements. Four units were fixed on tripods positioned on the anchor points and one unit was attached to the pole as described in the previous section. PulsON P400 series TimeDomain UWBs were used [18]. These units can be operated in three different ranging modes. Firstly, they can be used as standalone devices that do not need to be linked with a computer. Secondly, they can be operated as peer-to-peer ranging devices and lastly as part of ranging networks such as RangeNet. RangeNet is a TimeDomain network designed for the usage of TW-ToF that also includes coarse range estimates (CREs) or filtered range estimates (FREs). Coarse range estimates are derived from the first arriving signal strengths at the corresponding unit. The CREs are regularly updated and are available in form of FREs. Next to ranging between two units, the estimation of the position of a unit using a Kalman filter is possible, as well as creating a motion model, computing tuning parameters and estimating geometric dilution of precision (GDOP) values.

### 4.3. Employed Processing Software

Following the completion of the measurements, the raw data were post-processed. In the case of the GNSS measurements, RTKLib was used. RTKLib is an open-source toolkit created to calculate standard and precise positions. Using raw data as input, the data can be processed in real-time or in a post-processing environment to determine accurate positions. Positioning can be performed either with a reference station, precise point positioning (PPP), precise ephemerides or clock data. All the major satellite systems are available for processing the data. Post-processing of the data was performed using the RTKPost tool from the toolkit. With this tool, a wide range of settings can be adjusted to estimate the positions.

### 4.4. Determination of the Ground Truth

Ground truth coordinates of all points (i.e., anchor and grid points) were determined to be able to assess the accuracy of the estimated grid point coordinates as well as the accuracy of the measured ranges. For this, static GNSS observations with the Spectra SP80 geodetic receiver were carried out. The GNSS static measurements were post-processed with the precise satellite ephemeris and satellite clock corrections from the IGS. To further improve the ground truth accuracy, a reference station (i.e., base) was set up on a pillar on the roof of the Electrotechnical Institute building of the Vienna University of Technology. Base station data were used to correct the GNSS measurements collected on Karlsplatz Square. Using Equation (Equation 17), the ground truth ranges between anchor points and grid points were calculated.

## 5. Results

The major aim of this investigation was to evaluate the positioning in GNSS-challenged environments [28] when GNSS and UWB measurements are fused with a WLS approach, as shown in Section 3.3. This section presents the ranging and positioning performance of the proposed approach for data collected as shown in Section 4.

### 5.1. Ranging Assessment

In this section, the ranging performance of UWB, GNSS and their combination in terms of accuracy and precision is investigated.

#### 5.1.1. Evaluation of UWB Ranging Data

Figure 11 presents time series of all UWB range measurements. All UWB range observations from all seven grid points (G1–G7) to four different anchor points, i.e., P20, P18, P19 and P07, are shown. The different colors of the ranges in the time series represent the time at which the data were collected at a specific grid point, as indicated in the legend. The white points show only range observations that were filtered as outliers. By using the median of the range measurements per grid point, a buffer around the median was defined, and any measurements outside of this buffer were excluded as outliers.

Average ranging error, median ranging error, maximum, minimum and standard deviation σ were calculated for UWB measurements. The errors were calculated based on the distances calculated from the ground truth coordinates of all points and their differences from the measured UWB ranges. Table 1 summarizes the results for all seven grid points. Empty rows, such as for grid points G4 and G5, indicate that no measurements to the respective anchor point were available and/or usable. The ranging accuracy was generally at the decimeter-level with some exceptions, such as measurements on grid point G7, where it was above 1 m. The standard deviations were in the range of a few centimeters. The most precise measurements showed standard deviations of around 1 cm, whereas the most imprecise measurements showed standard deviations of around 18 cm. The worst standard deviations occurred on grid points with many outliers and/or missing data. Due to some problems with the connection between UWB units, the data for grid points G4–G6 could reflect this situation. For grid point G7, the results show larger deviations because of the geometry within the network and obstruction of LoS due to the tree that is right next to this point. Additional reasons why data quality might not be satisfying in some cases could be that the surveyor had to hold the pole the entire time and small movements of it could have occurred due to windy conditions. Furthermore, the surveyor could have broken the LoS between the UWB units when holding the pole, which probably resulted in a loss of signal strength. As mentioned in Section 2.2, UWB units operate with frequencies in a range 3.1 to 10.6 GHz, whereas the low frequencies enable the signal to pass through obstructions [9]. Even though it is possible for the signal to penetrate obstructions, the measurement quality can still be affected. Similarly, the presence of pedestrians also resulted in the loss of signal strength.

Figure 12 shows a visualization of the average range errors before and after the UWB-only WLS approach. The range error deviationsare shown on the *y*-axis. The *x*-axis relates every range measurement to an anchor point. The deviations of the range errors are shown with pairs of box plots. The left box plot in each subplot shows the deviations before WLS (i.e., raw uncorrected measurements) while the right box plot shows deviations after WLS (i.e., adjusted measurements). Most of the deviations decreased, which is represented by the narrow interquartile ranges (i.e., boxes). The median value is represented by the red line in the interquartile range. Outliers are represented by blue crosses. Most of the range error medians were improved after the adjustment (i.e., closer to 0 m error on the *y*-axis).

There are at least three range measurements available where an improvement in accuracy can be observed following the adjustment. In cases where only two or even only one range measurement is available during the data collection, the improvement is not significant if it can be observed. In the case of grid point G7, even though there are three or more ranges available, an improvement cannot be observed.

#### 5.1.2. Evaluation of GNSS Ranging Data

This section focuses on the GNSS data and the evaluation of the GNSS-only WLS approach. As mentioned in Section 3.3.2, baselines had to be calculated first.

A similar figure as for the UWB-only measurements was produced for the GNSS-only measurements (see Figure 13). There are no outages and there are as many baselines available as there were collected grid point data. It can be seen in Figure 13 that GNSS baseline errors between the anchor points and grid points G2, G3 and G4 are small and close to 0 m (within 2 cm from the ground truth). The reason for this can be explained by good satellite geometry and little to no obstructions of LoS. Grid points G1, G5, G6 and G7 show GNSS baseline errors of 5 cm to 20 cm. Despite the lower accuracy, the precision at G1, G5 and G6 grid points is similar to the other points and it is within the decimeter level. Because the baselines are dependent on the GNSS data that were processed with broadcast ephemerides, such deviations are expected. As for G7, the unsatisfactory data quality is most likely due to the location of the point, which is right next to a large tree that obstructed the clear sky as shown in Figure 9.

The GNSS-only WLS shows no significant change in the baseline accuracy.

#### 5.1.3. Evaluation of GNSS and UWB Fused Range Data

This section presents the evaluation of the multi-sensor WLS. The functional model for this calculation method is as shown in Equation (Equation 20). As described in Section 3.3.3, the first part of the model consists of the model representing the UWB measurements and the second part consists of the model using only GNSS data. The evaluation of the range data was performed similarly as in the previous Section 5.1.1 and Section 5.1.2.

The results indicate improvements for the fused UWB and GNSS approach. Grid point G1 showed the biggest increase in accuracy of more than 10 cm for the distances to anchor points P18 and P20. The standard deviation for all ranges to G1 has also been reduced to around 1 to 2 cm. The multi-sensor model also had an impact on grid points G2, G3 and G4.It seems that addition of the GNSS model negatively impacted the UWB ranges for grid point G5. A ranging performance improvement of a couple of centimeters was observed for grid point G6.A similar ranging improvement was observed for G7 even though the ranging accuracy still remained around 1 m.

The results show that the multi-sensor solution improves range accuracy by a few centimeters. The following section will show how this is reflected in the positioning solution.

### 5.2. Positioning Performance

This section covers the assessment of the positioning performance of the measured grid points for all three approaches, i.e., UWB and GNSS-only solutions and the multi-sensor solution.

All deviations from the ground truth (i.e., positioning errors or positioning accuracy) were calculated with Equation (Equation 21). GT denotes the ground truth coordinates of the corresponding grid point $G_{i}$ and *dev* describes the calculated positioning error. ${∥\cdot ∥}_{2}$ is the notation for the Euclidean norm. GT and $G_{i}$ are both vectors in the form [x,y,z]${}^{T}$.
(21)$$\begin{array}{c} dev={∥GT-G_{i}∥}_{2},i=1,\cdots ,7 \end{array}$$

#### 5.2.1. Positioning with UWB-Only WLS

As mentioned in Section 5.1.1, the measured and uncorrected GNSS coordinates were used as an approximate position. A comparison of the estimated (i.e., adjusted) grid points (red markers), measured grid points (light blue markers) and the ground truth (yellow markers) is shown in Figure 14. The points marked with black triangles represent the anchor points (P19, P20, P07’, P18’), while the blue points show the measured grid points (G1–G7). The notations $G_{i},\left(i=1,2,\cdots ,7\right)$ refer to the grid points and $GT_{i},\left(i=1,2,\cdots ,7\right)$ denotes the ground truth for the respective grid point.

Table 2 shows the UWB-only WLS positioning performance at all grid points. The table shows the average error, median error, maximum and minimum errors and standard deviation. Points G1, G2 and G3 achieve sub-meter positioning accuracy with high precision (i.e., low standard deviation). Points G4 and G5 show a significant increase in the average positioning error to 4 m and almost 8 m, respectively. This is most likely due to the unavailability of the UWB ranges as shown in Table 1. Only measurements from two anchor points were available for G4 and only one for the grid point G5. The position estimate for the G6 grid point is within the sub-meter level, most likely due to the availability of measurements from three anchor points (see Table 1). Measurements from all anchor points were available at G7. Nevertheless, the achieved accuracy was on average almost 2 m with the highest standard deviation of almost 0.3 m. There are two factors that probably caused this decrease in performance compared to points G1, G2 and G3. The first factor relates to the network geometry where all anchor points are south or southeast from G7 (see Figure 10). The second factor may relate to the obstacles close to the point, such as the tree (explained in Section 5.1.1).

#### 5.2.2. Positioning with GNSS-Only WLS

In the following, the results for the measured GNSS grid points with the GNSS-only WLS approach are assessed. Figure 15 shows the GNSS measurement data after being post-processed in RTKLib (introduced in Section 4.3) and filtered with the self-developed clustering method that was described in Section 3.1. The points marked with black triangles represent the anchor points (P19, P20, P07’, P18’) while the blue points show the measured grid points (G1–G7). All points were measured in ECEF (Earth-centered Earth-fixed coordinate frame). The *x*- and *y*-axis were both adjusted towards the origin for better readability. Grid points G1–G6 were measured with little to no obstruction in LoS to the satellites. However, G7 was measured close to the large tree. Because of that, the observation results of G7 show larger errors and lower precision.

Table 3 shows the simple statistics calculated for errors in the measured GNSS data at each grid point. The data indicate centimeter-level accuracies for grid points G1, G2, G3 and G4 because there are almost no obstructions in the surroundings of each of these points. Another reason for this result could be the large number of available satellites, which was about 17 to 18, and consequently, a good satellite–receiver geometry. The number of visible satellites was the lowest (12 satellites) at grid point G7, where the clear sky was obstructed due to the large tree (see Figure 9). The low accuracy for G5 and G6 could be explained through similar reasoning as in Section 5.1.2. By trying to hold the pole still in the windy environment, the surveyor might have caused these large positioning errors.

#### 5.2.3. Positioning with GNSS/UWB WLS

This section shows the assessment of the positioning performance of the GNSS/UWB WLS approach introduced in Section 3.3.3. The visualization of these results is shown in Figure 16.

The addition of the GNSS baseline measurements improved the average UWB-only positioning performance for points G1, G2 and G3 (0.3 m, 0.9 m and 0.5 m, respectively). On these points, GNSS performance was at centimeter-level due to the open-sky environments. A decimeter-level performance was achieved at these points with the multi-sensor solution. The biggest improvement can be observed for points G4 and G5. As previously mentioned, a maximum of two UWB measurements were available at these points, which resulted in the accuracy of 4 and 7.8 m for the UWB-only solution. With the addition of more accurate GNSS baseline measurements, the accuracy was increased to half-a-meter level. The performances of both the UWB-only and the GNSS-only solution on point G6 were similar (1.05 m and 1.01 m, respectively). This resulted in the improved positioning 0.99 m accuracy of the multi-sensor solution. This is most likely due to the increase in redundant measurements and the improved network geometry. A decrease in accuracy of about 90 cm can be noted for grid point G7. The reason for this is most likely due to the fact that G7 is located outside of the area that all anchor points enclose. By being outside of this anchor point area, all range measurements come from similar directions and therefore result in poor geometry. This influences the ranges significantly and consequently makes them longer than they are in reality. With these incorrect ranges, the current positioning solution was reached. The numerical statistical results are presented in Table 4.

## 6. Concluding Remarks and Outlook

This study has shown some of the possible ranging and positioning effects of GNSS/ UWB fusion and compared the results with those of UWB-only and GNSS-only methods. The methods used for the analysis include a DBSCAN clustering approach and WLS algorithm to acquire the user’s static position. The experimental results show the relevant data acquisition process and the related analysis results.

The ranging and positioning solution performance with the WLS approach for the integration of GNSS and UWB ranging has been discussed. Positioning with GNSS has limits when it comes to challenging environments and transition from out- to indoors and vice versa. With the LoS of the signal path between satellites and the receiver on Earth obstructed, several (deci-)meters of positioning error can be observed. To address this problem, a combined solution of GNSS positioning with UWB ranging was introduced. UWB ranging is a state-of-the-art ranging and positioning method using TW-ToF technology between two transceivers. By positioning one UWB unit at a fixed and known location as an anchor point, it is possible to estimate the distance between this anchor point unit and the unit used at the location to be determined. By sending out signals in a bandwidth of 3.1 to 10.6 GHz from the main unit at the sought-after location to the anchor node, which are then sent back to the transmitter, it is possible to determine ranges with cm-level accuracy.

The measurement campaign was carried out in an environment that includes GNSS-challenging situations (e.g., obstruction of LoS). Four UWB anchor points were set up. On each anchor point, one UWB TimeDomain transceiver was placed. To determine approximate coordinates and ranges for each grid point, a multi-GNSS receiver and the UWB unit were placed on a measuring pole. The data post-processing was performed with the open-source software package RTKLib.

The ranging and positioning results show definite improvements in accuracy to a predetermined ground truth. The results of a UWB-only approach resulted in minor improvements in accuracy for most grid points when at least three range measurements were available. When fewer than three ranges between the anchors and grid points were available, the results show a significant deterioration of accuracy (up to several meters). Compared to that, the GNSS-only solution showed almost no changes. However, the combined solution of the introduced GNSS/UWB-fusion WLS approach supported the grid point position estimates where fewer than three UWB ranges were available the most. By producing redundancy and filling gaps for the missing range measurements, improvements of several meters were achieved, reducing the position error to the dm-range. Additionally, there was one grid point outside of the area that was enclosed by the anchor points. This grid point experienced a rather large decrease in accuracy, most likely due to poor geometry.

For future field work, it should be considered that ranging and positioning of points should be conducted within the borders of the mentioned anchor points and possibly with a larger number of anchor points. Additionally, considering the windy weather situation in which the measurements were performed, proper equipment should be prepared to counter for these situations to reduce the influence of the user on the measurements. Furthermore, to compare different positioning methods an extended Kalman filter could be implemented to see if similar or even better results can be achieved. If smartphones are the platform used, the embedded inertial sensors [29], i.e., accelerometers and gyroscopes together with magnetometers, can be utilized, leading to a continuous ubiquitous positioning solution.

UWB is being more widely adopted due to its robustness against multiple paths and its centimeter-level accuracy [30]. Some newer smartphones (e.g., Samsung Galaxy S21, Apple iPhone 11 series) are already equipped with low-cost and small UWB chip-sets. UWB chip-sets in mobile devices have not yet been used for ranging applications. Due to the inclusion of UWB chip-sets in mobile devices, this possibility arises. Furthermore, UWB technology is the foundation of tracking tags like Apple’s AirTag and Samsung’s SmartTag Plus [31]. These tags let you unlock your car or home’s front door as you approach them with your phone. This is just the beginning of the utilization of UWB for localization in general. The European Telecommunications Standards Institute (ETSI) [32] confirms that UWB may become the standard for indoor positioning in the future.

A main drawback of UWB in a smartphone is that, in order to achieve a short pulse width, the UWB device has a high power consumption for a single packet transmission [33,34]. Hence, using the TW-ToF protocol where multiple packets have to be exchanged increases the energy consumption.

For maintaining overall performance for a number of applications, the fusion of multiple positioning technologies and sensors is a necessity. GNSS-only solutions are difficult or even impossible to implement in urban canyons or NLoS conditions. In these cases, multi-path effects and a reduction in the number of satellites in view may cause large positioning errors or even failure. The gap in satellite coverage or GNSS performance is not acceptable for many applications and has to be addressed by using complementary technologies, such as UWB technology.

## Figures and Tables

**Figure 1 sensors-23-03303-f001:**
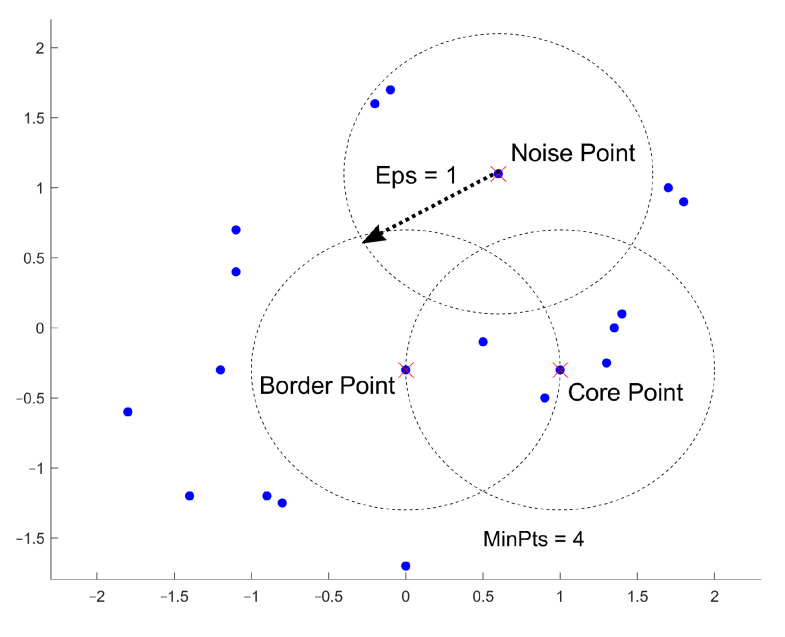
Definition of core, border and noise points.

**Figure 2 sensors-23-03303-f002:**
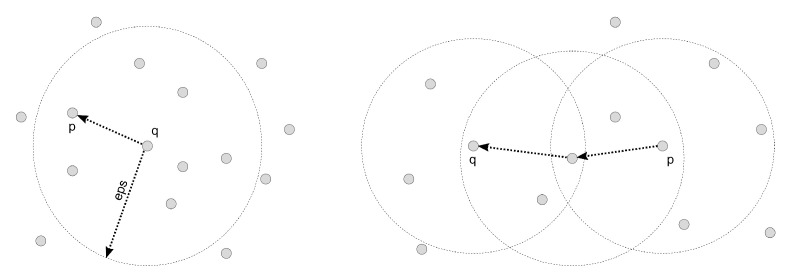
Definition of density-reachable points and density-connected points.

**Figure 3 sensors-23-03303-f003:**
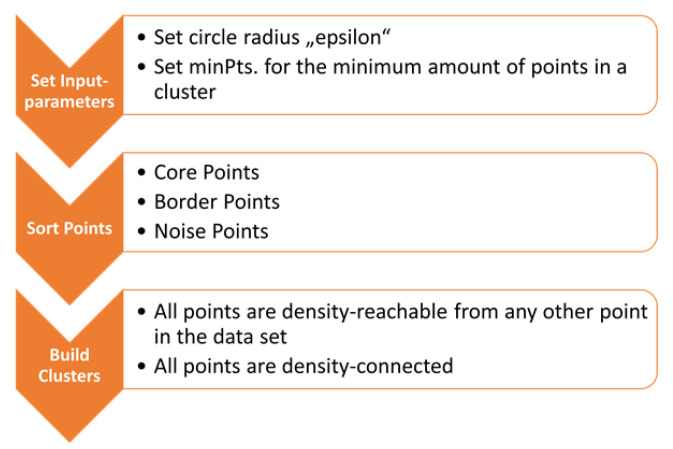
Flowchart of the density-based spatial clustering of applications with noise (DBSCAN).

**Figure 4 sensors-23-03303-f004:**
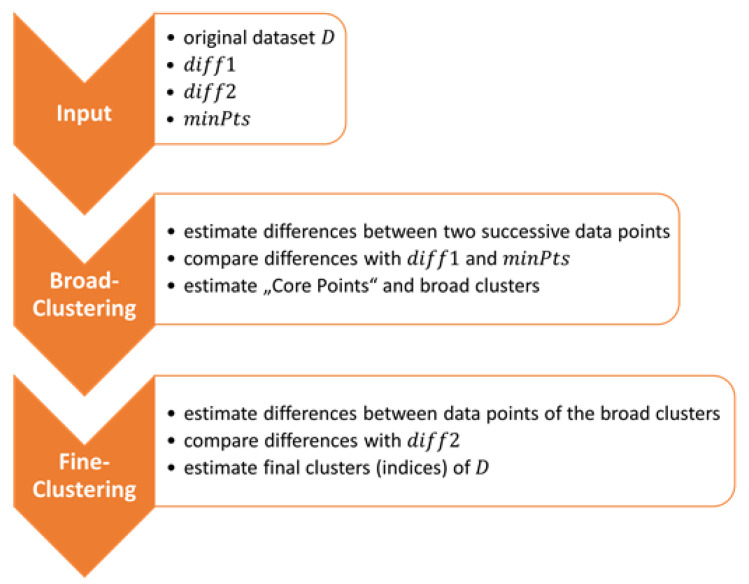
Flowchart of the DBSCAN-derived clustering method.

**Figure 5 sensors-23-03303-f005:**
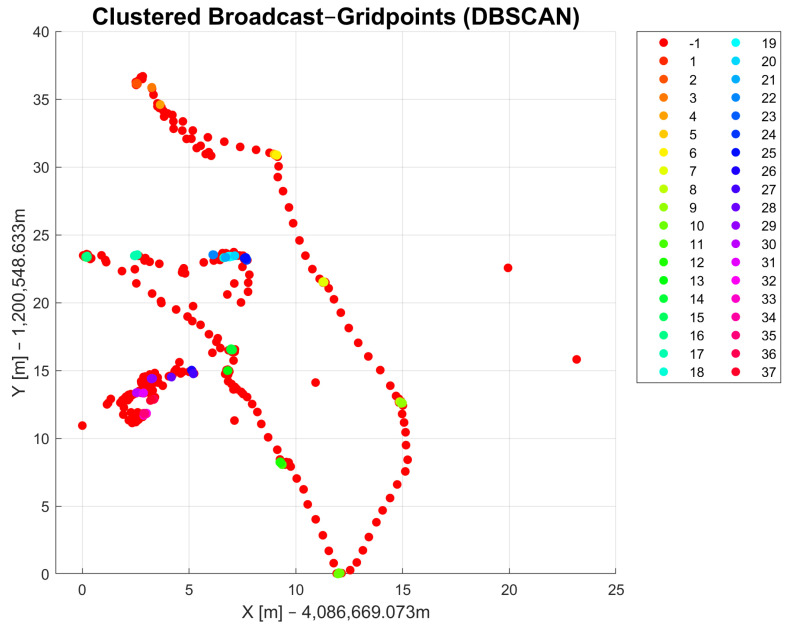
Results of DBSCAN clustering.

**Figure 6 sensors-23-03303-f006:**
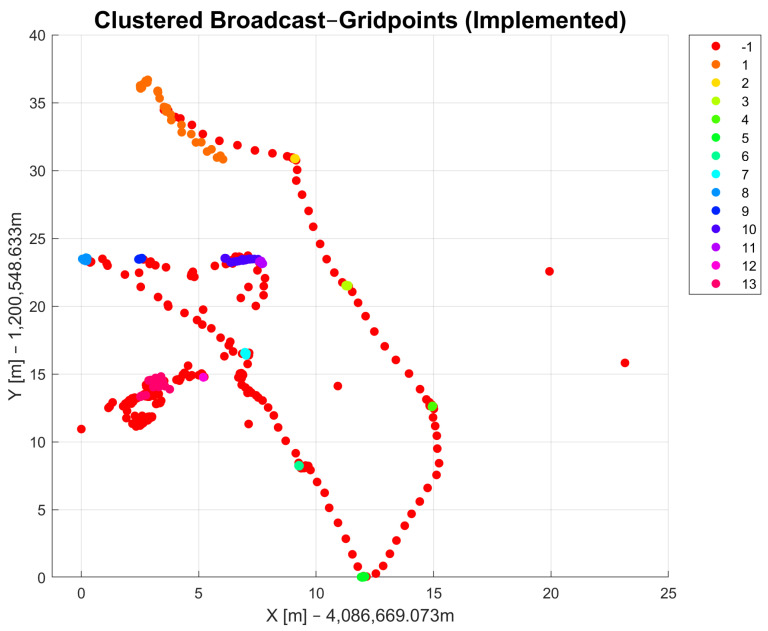
Results of DBSCAN-derived clustering method.

**Figure 7 sensors-23-03303-f007:**
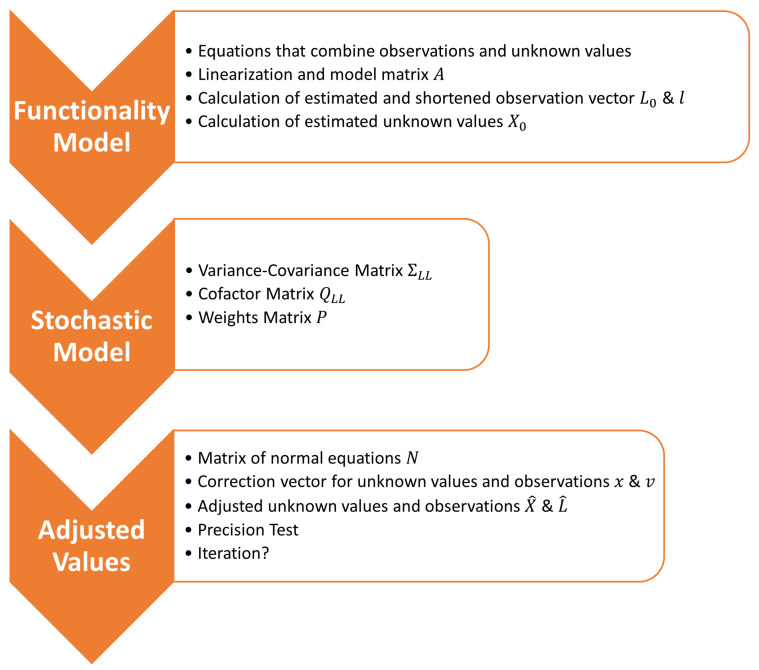
Flowchart of weighted least squares (WLS).

**Figure 8 sensors-23-03303-f008:**
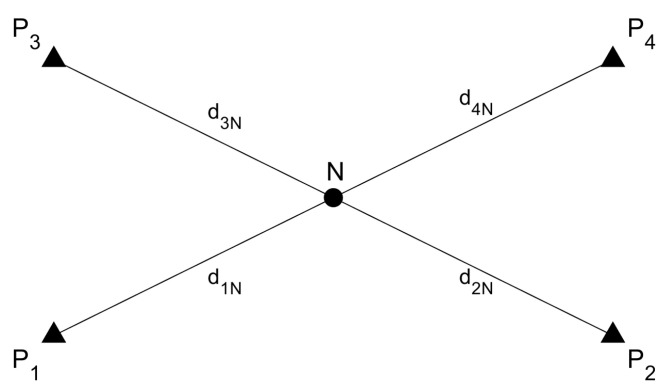
Four anchor points ($P_{1}$, $P_{2}$, $P_{3}$ and $P_{4}$) and one grid point (*N*).

**Figure 9 sensors-23-03303-f009:**
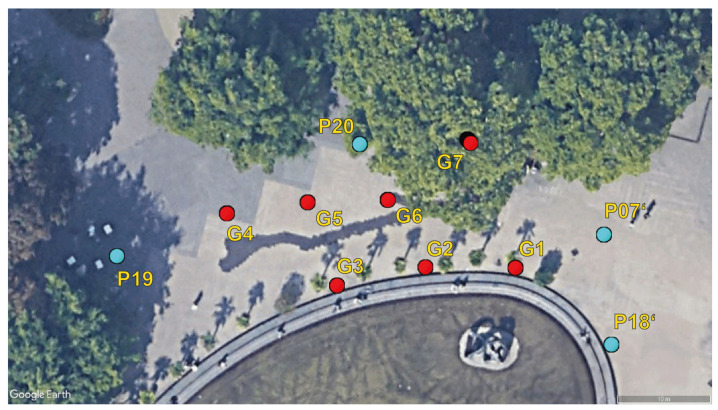
Survey area in a park.

**Figure 10 sensors-23-03303-f010:**
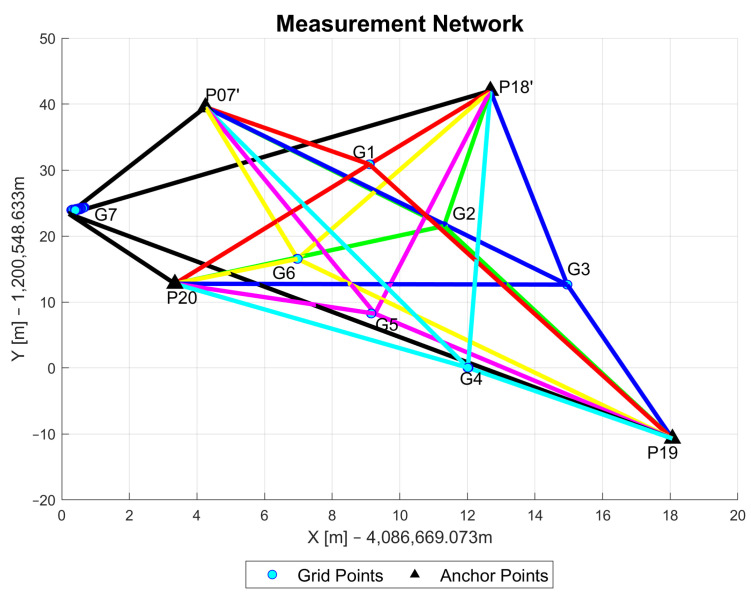
Sketch of the test site showing between which grid point and anchor point UWB range measurements were collected.

**Figure 11 sensors-23-03303-f011:**
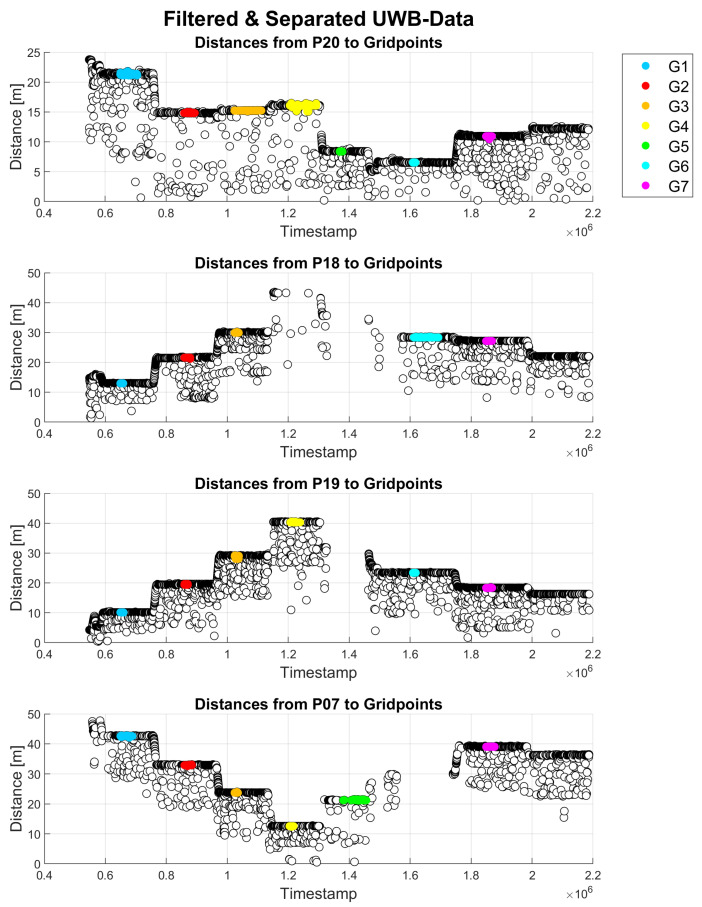
UWB range measurement between anchor points (P20, P18, P19 and P07) and all grid points as indicated in the legend.

**Figure 12 sensors-23-03303-f012:**
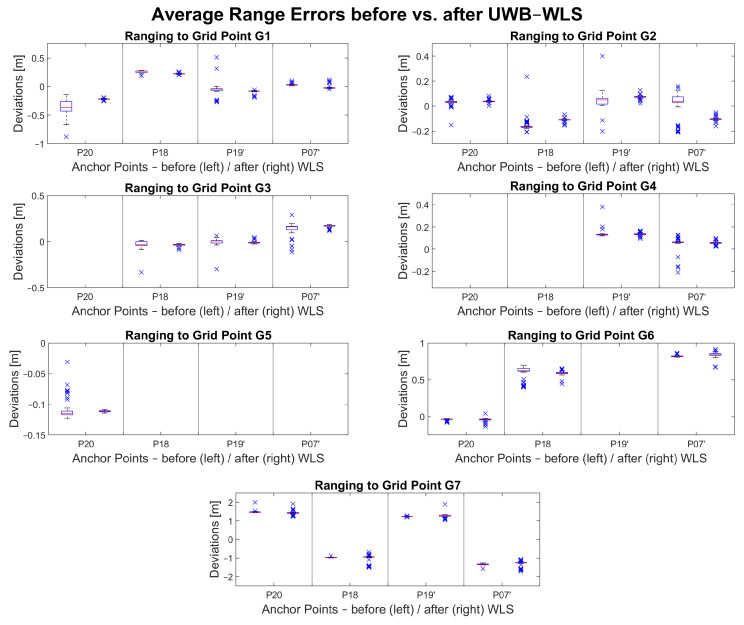
UWB range errors before and after UWB-only WLS adjustment.

**Figure 13 sensors-23-03303-f013:**
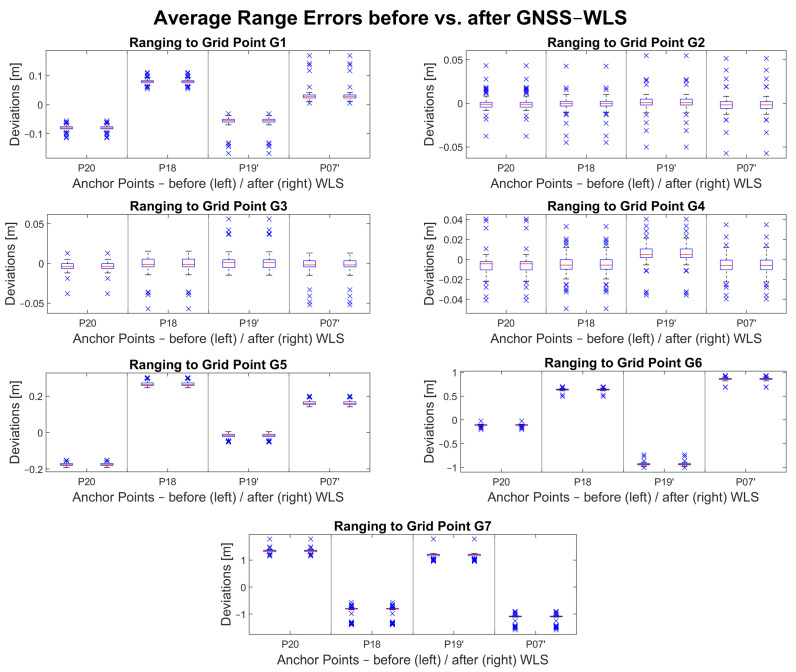
GNSS baseline errors before and after GNSS-only WLS adjustment.

**Figure 14 sensors-23-03303-f014:**
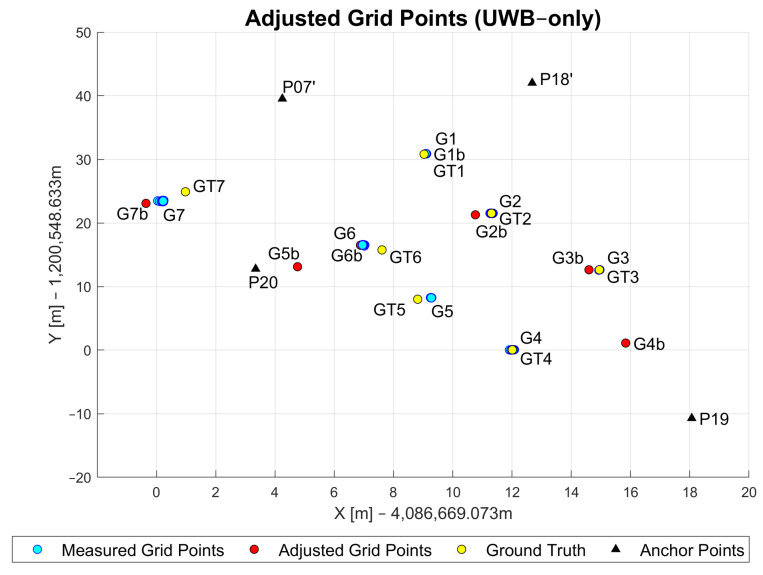
Adjusted, measured and ground truth grid point coordinates based on UWB-only approach.

**Figure 15 sensors-23-03303-f015:**
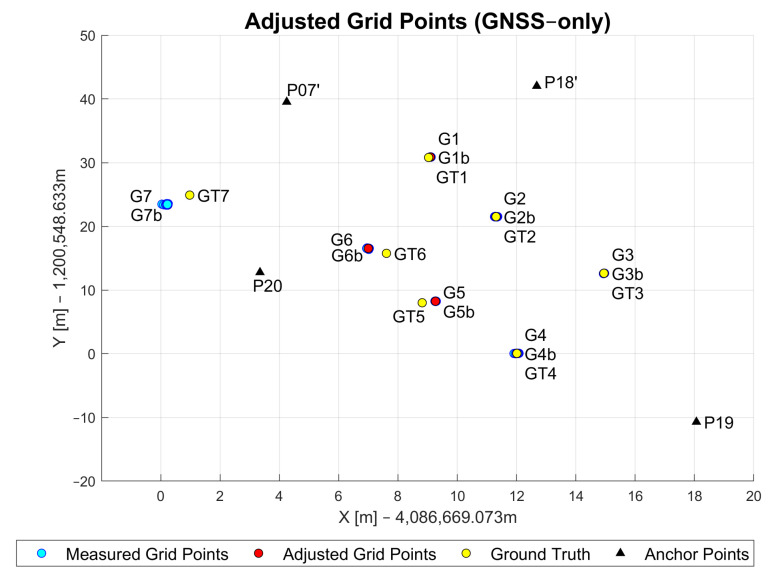
Adjusted, measured and ground truth grid point coordinates based on GNSS-only approach.

**Figure 16 sensors-23-03303-f016:**
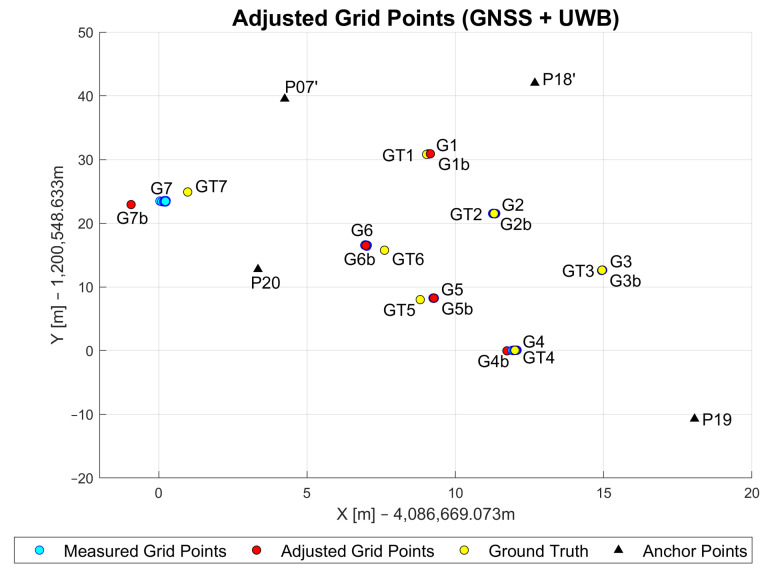
Adjusted, measured and ground truth grid point coordinates based on GNSS/UWB approach.

**Table 1 sensors-23-03303-t001:** Measured UWB range errors in relation to the ground truth in (m).

GP No.	AP No.	Avg.	Median	Max.	Min.	σ
G1	P20	−0.352	−0.367	−0.143	−0.880	0.123
	P18	0.259	0.268	0.280	0.190	0.020
	P19’	−0.048	−0.045	0.509	−0.273	0.085
	P07’	0.032	0.030	0.097	0.012	0.014
G2	P20	0.030	0.034	0.073	−0.152	0.030
	P18	−0.157	−0.167	0.235	−0.209	0.046
	P19’	0.050	0.050	0.401	−0.201	0.058
	P07’	0.014	0.039	0.159	−0.211	0.096
G3	P20					
	P18	−0.030	−0.032	0.013	−0.333	0.042
	P19’	−0.002	0.007	0.064	−0.299	0.036
	P07’	0.169	0.161	1.091	−0.114	0.177
G4	P20					
	P18					
	P19’	0.133	0.130	0.378	0.118	0.028
	P07’	0.056	0.063	0.130	−0.210	0.048
G5	P20	−0.111	−0.115	−0.031	−0.123	0.013
	P18					
	P19’					
	P07’					
G6	P20	−0.039	−0.035	−0.025	−0.078	0.011
	P18	0.617	0.624	0.697	0.400	0.066
	P19’					
	P07’	0.824	0.818	0.862	0.808	0.015
G7	P20	1.473	1.461	1.987	1.443	0.058
	P18	−0.968	−0.969	−0.994	−0.880	0.018
	P19’	1.236	1.233	1.285	1.192	0.016
	P07’	−1.344	−1.349	−1.576	−1.269	0.035

**Table 2 sensors-23-03303-t002:** Adjusted UWB-only positioning error statistics (m).

PNo.	Avg.	Median	Max.	Min.	σ
G1	0.317	0.317	0.353	0.312	0.034
G2	0.850	0.849	0.900	0.804	0.019
G3	0.513	0.513	0.532	0.484	0.013
G4	4.024	4.024	4.102	3.917	0.023
G5	7.800	7.797	7.819	7.795	0.014
G6	1.049	1.054	1.008	0.972	0.046
G7	2.693	2.797	3.034	1.807	0.396

**Table 3 sensors-23-03303-t003:** Measured GNSS grid point error statistics in (m).

PNo.	Avg.	Median	Max.	Min.	σ
G1	0.095	0.096	0.271	0.267	0.040
G2	0.002	0.002	0.089	0.072	0.015
G3	0.009	0.008	1.007	0.132	0.084
G4	0.008	0.008	0.140	0.110	0.022
G5	0.515	0.505	0.891	0.432	0.072
G6	1.006	1.007	0.987	0.975	0.034
G7	1.941	1.988	2.171	1.439	0.290

**Table 4 sensors-23-03303-t004:** GNSS/UWB-fusion error statistics in (m).

PNo.	Avg.	Median	Max.	Min.	σ
G1	0.161	0.165	0.077	0.234	0.034
G2	0.076	0.075	0.134	0.063	0.019
G3	0.016	0.015	0.071	0.043	0.013
G4	0.437	0.437	0.514	0.391	0.023
G5	0.516	0.515	0.490	0.545	0.014
G6	0.985	0.989	0.896	0.958	0.046
G7	3.574	3.693	3.911	2.527	0.396

## Data Availability

The data can be made available upon request.

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
