# Peer review of "Fusion of GNSS Pseudoranges with UWB Ranges Based on Clustering and Weighted Least Squares"

_sensors, 2023, doi:10.3390/s23063303_

Round 1

Reviewer 1 Report

This manuscript shows in detail some of the possible ranging accuracy, positioning effects of GNSS/UWB fusion positioning and compares the results of ranging and positioning with those of UWB-only and GNSS-only methods. The methods used for the analysis include a density clustering based approach (DBSCAN) and WLS algorithm to acquire the user's static position. The experimental results show in detail the relevant data acquisition process and the related analysis results. I acknowledge that these contents have some reference value or significance. However, the related research methods are mature and the manuscript does not contain enough innovative contents. In addition, the writing of the manuscript is not concise enough, the textual expressions are heavy and redundant, and there are few relevant references in the introduction.

Author Response

Dear Reviewer,

our reply is given in the attached document.

Best regards,

Guenther Retscher

Reviewer 2 Report

Corrections are given in the PDF document.

Author Response

(The authors gave the same response as above.)

Reviewer 3 Report

1. In the Abstract, the authors should explain how clustering is used in the paper. Why need to combine GNSS and UWB, as GNSS is for outdoor environment and UWB is usually used for indoor environment. What is the application of the proposed system?

2. In the abstract, the authors mentioned “GNSS Stop-and-Go measurements were conducted simultaneously to UWB range observations from given anchor points”. It is not clear what do you mean by this.

3. DBSCAN clustering is described in the paper. What is the connection to the localization problem in problem addressed in the paper.

4. The size of the figures (for example figure 3 and figure 4) are too large, please reduce the size of the figures.

5. In Figure 5 and Figure 6, what is the meaning of the points. Are you showing the localization results? The Figure 5 and Figure 6 should be moved to correct pages.

6. Section 3.3 is about sensor fusion, the authors need to highlight what kind of sensors are fused? The process of sensor fusion is not shown clearly. Are you using WLS to fuse UWB and GNSS measures? Or WLS is just used to filter the raw ranging measurement.

7. It is not clear how to localize with UWB and how to localize with GNSS. Are you using the same technique. How you know the position of GNSS anchors?

8. In Figure 10, which one is UWB anchor and which one is GNSS anchor? Please make it clear in the figure.

9. It is not easy to follow Figure 11, what kind of ranging information are showing there?

10. Use some tables to summarize the localization results, so people can compare the performance of different approaches easily.

11. I suggest the authors to look at the following paper, as they are quite relevant to the topic addressed here.
Distributed ranging slam for multiple robots with Ultra-wideband and odometry measurements. IEEE IROS 2022.

Author Response

(The authors gave the same response as above.)

Round 2

Reviewer 1 Report

This manuscript discussed the combined positioning of UWB and GNSS, and it has some certain reference value. In general, the expression is very lengthy and redundant. It is recommended to further simplify, especially about the GNSS algorithm section and the examples section. Moreover some places are not clear, this manuscript only use the GNSS baseline length, and this should be emphasized. Such as figure 12-16,about these deviations of ranges and positioning can be deleted. Figure 10 using xy of the cartesian coordinates is not suitable, should be replaced by plane coordinate system.

Author Response

Dear Reviewer,

please see our reply in the attached document.

Best regards,

Guenther Retscher

Reviewer 3 Report

Thanks for providing a thorough correction of the paper. All my comments are well addressed.

Author Response

Dear Reviewer,

thank you very much for your time and confirmation that we had addressed your comments after the first round of your review.

Best regards,

Guenther Retscher